Detours in long-distance migration across the Qinghai-Tibetan Plateau: individual consistency and habitat associations

Liu Dongping dpliu@caf.ac.cn
Zhang Guogang zm7672@126.com
Jiang Hongxing
Lu Jun
Key Laboratory of Forest Protection of State Forestry Administration, Research Institute of Forest Ecology and Environment Protection, Chinese Academy of Forestry , Beijing , China
Harrison Xavier
Electronic publication date: 2018 Jan 31
Publication date: 2018
Volume: 6
Electronic Location ID: e4304
Received 2017 Jul 19; Accepted 2018 Jan 10
Copyright: ©2018 Liu et al.
Copyright year: 2018
Copyright holder: Liu et al.
License: This is an open access article distributed under the terms of the Creative Commons Attribution License, which permits unrestricted use, distribution, reproduction and adaptation in any medium and for any purpose provided that it is properly attributed. For attribution, the original author(s), title, publication source (PeerJ) and either DOI or URL of the article must be cited.
License URL: https://creativecommons.org/licenses/by/4.0/

Keywords: Pallas’s Gull, Satellite tracking, Detour, Qinghai-Tibetan Plateau, Ecological barrier, Larus ichthyaetus, Migration strategy, Loop migration, Repeatability, Consistency

Funding: National Key Research and Development Program of China 2016YFC1201601 Key Scientific and Technological Project of China 2004BA519A63 State Forestry Administration of China This study was financially supported by the National Key Research and Development Program of China (2016YFC1201601), the Key Scientific and Technological Project of China (2004BA519A63) and the Wildlife Surveillance Program from the State Forestry Administration of China. The funders had no role in study design, data collection and analysis, decision to publish, or preparation of the manuscript.

==============================
Migratory birds often follow detours when confronted with ecological barriers, and understanding the extent and the underlying drivers of such detours can provide important insights into the associated cost to the annual energy budget and the migration strategies. The Qinghai-Tibetan Plateau is the most daunting geographical barrier for migratory birds because the partial pressure of oxygen is dramatically reduced and flight costs greatly increase. We analyzed the repeated migration detours and habitat associations of four Pallas’s Gulls Larus ichthyaetus across the Qinghai-Tibetan Plateau over 22 migration seasons. Gulls exhibited notable detours, with the maximum distance being more than double that of the expected shortest route, that extended rather than reduced the passage across the plateau. The extent of longitudinal detours significantly increased with latitude, and detours were longer in autumn than in spring. Compared with the expected shortest routes, proximity to water bodies increased along autumn migration routes, but detour-habitat associations were weak along spring migration routes. Thus, habitat availability was likely one, but not the only, factor shaping the extent of detours, and migration routes were determined by different mechanisms between seasons. Significant between-individual variation but high individual consistency in migration timing and routes were revealed in both seasons, indicating a stronger influence of endogenous schedules than local environmental conditions. Gulls may benefit from repeated use of familiar routes and stopover sites, which may be particularly significant in the challenging environment of the Qinghai-Tibetan Plateau.

Introduction

Migratory birds often follow detours, which are spatial deviations from the most direct route, to minimize energy expenditure or travel time and ultimately arrive at their destinations safely and optimally (Alerstam, 2001; Alerstam, 2011). Although birds travel longer distances and incur extra energetic costs with migration detours, they can be sufficiently compensated by reducing the risk of disturbance and predation (Klaassen et al., 2006; Ydenberg, Butler & Lank, 2007), reducing costs in time to fuel in high-quality stopover sites (Lindström et al., 2011; Aharon-Rotman et al., 2016), or reducing energetic costs to fly in favourable wind conditions (Vansteelant et al., 2017; Tøttrup et al., 2017). When confronted with a heterogeneous environment regarding to topography, weather condition and food availability, long-distance migrants might have a great capacity to follow variable degree of detours that are characterized by a combination of favorable factors (Hahn et al., 2014). Analysing the extent of detours as well as the environmental conditions experienced en route is crucial for determining why birds employ such migration strategies and the underlying drivers.

Mountain ranges, oceans, deserts, ice fields, and bad-weather fronts constitute the main ecological barriers to migratory birds (Berthold, 2001). To accomplish a non-stop, long-distance flight over an extensive barrier, migratory birds require heavy fuel loads, whose transport incur extra energetic costs (Gill et al., 2009). Therefore, to minimize energetic expenditures, birds usually take a detour in the optimum route that includes a shortcut across part of the barrier (Alerstam, 2001). Previous studies have primarily focused on migration detours across ecological barriers such as deserts, oceans and bad weather fronts (Mellone et al., 2011; Hawkes et al., 2013; Smolinsky et al., 2013; Hahn et al., 2014; Vansteelant et al., 2017), but the performance of migrants confronted with the challenging environment of high plateaus remains poorly explored, although this type of migration has been documented in a few species (Zhang et al., 2011b; Zhang et al., 2014; Liu et al., 2012; Hawkes et al., 2013). Compared to travel over oceans and deserts, birds migrating over high plateaus must overcome the disadvantage of a reduced partial pressure of oxygen by increasing their metabolic rates (Scott, 2011; Hawkes et al., 2013), and spend more energy during flapping flight due to the reduced lift generation in thinner air (Pennycuick, 2008). However, birds can make stopovers on plateaus when necessary, and take advantage of the predictable daily slope winds to climb high plateaus (Hawkes et al., 2011). Consequently, this may result in different detour patterns among migratory birds encountering high plateaus.

Endogenous programmes largely control the migration schedule (Berthold, 1996), but environmental conditions (e.g., climate conditions and food availability) also greatly affect migration flexibility, especially at the individual level (Both, 2010; Studds & Marra, 2011). Generally, strong endogenous control results in lower within- (i.e., individual consistency) than between-individual variation in migration schedules over successive years, whereas environmental conditions have the opposite effect (Vardanis et al., 2011; López-López, García-Ripollés & Urios, 2014). Therefore, analysing the variation in repeated migration tracks taken by long-distance migrants is an important way to assess the degree of flexibility in migration and understand the adaptive migration strategies under environment pressures.

In this study, we analysed the long-distance migration in Pallas’s Gull Larus ichthyaetus between Qinghai Lake in northwestern China and Bangladesh Bay as revealed by satellite tracking. En route, the gulls were confronted with the geographical barrier of the Qinghai-Tibetan Plateau (hereafter, QTP), which is the world’s highest and largest plateau with an average elevation exceeding 4,500 m and an area of 2,500,000 km2 (Zhang, Li & Zheng, 2002). It is a great challenge for birds to migrate at extremely high altitudes because the partial pressure of oxygen is dramatically reduced, which greatly increases flight costs (Pennycuick, 2008; Hawkes et al., 2013; Parr et al., 2017), so the gulls were expected to follow detours to minimize exposure to inhospitable environments on the ground. We analysed the spatial patterns of the detours and compared the differences in migration routes and timing between autumn and spring to test for variations in seasonal migration strategies. All individuals were tracked during consecutive years (from two to four years); so we were able to compare between- and within-individual variation in migration routes and timing to assess the degree of repeatability in migration detours and schedules. We also compared habitat composition along the observed routes and the expected shortest routes to assess the relationship between migration detours and habitat availability.

Methods

Satellite tracking and location data processing

We tracked four Pallas’s Gulls (denoted G1 to G4) breeding at Qinghai Lake (99.36–100.77°E, 36.32–37.25°N; altitude = 3,200 m) in northwestern China by satellite telemetry (12 g PTT-100; Microwave Telemetry Inc., Columbia, MD, USA; duty cycle 8 h on/15 h off) during 2006–2009 (Zhang et al., 2014). We received the Doppler-derived PTT location information from the CLS/Service Argos satellite tracking system that reports 1-sigma error radii of 150–1,000 m to reflect the location accuracy of location classes (LC) 1, 2 and 3 (Argos, 1996); LC 0, A, B and Z are not assigned accuracy estimates but provide a large amount of relatively accurate data that can be used to analyse gross movements after being filtered (Soutullo et al., 2007). We used the Douglas Argos-Filter Algorithm (v. 8.50) to identify and remove implausible auxiliary Doppler locations based on the distance moved, movement rate, and turning angle (Douglas et al., 2012). Only locations recorded during complete migratory journeys (those that covered the entire distance between the breeding and wintering grounds) were included in this study.

Parameters for migration timing and routes

For each bird and migratory season, we recorded a set of parameters to describe migration timing and routes that included departure date, arrival date, duration, shortest distance (the great-circle distance between the breeding and wintering site), cumulative distance (the total great-circle distance of the migration segments) and the average migration speed (calculated as the cumulative distance/duration). We plotted telemetry locations in Google Earth Pro (version 7.1.8.3036; Google Inc., Mountain View, CA, USA) to determine migration timing parameters by visually inspecting the raw data and determining the habitat of the stopover sites. If the PTTs did not transmit continuously, we used the median date between the last point at the previous location and the first point of the new location to calculate timing. We plotted the telemetry locations in ArcView GIS (version 3.3; Environmental Systems Research, Redlands, CA, USA) and used the XTools extension to compute route distance. We also calculated the straightness index as the ratio between the shortest distance and the cumulative distance to roughly measure the degree of the migration detours.

To explore the spatial and temporal patterns of the migration of Pallas’s Gulls across the QTP, we recorded the parameters of trans-QTP duration, trans-QTP distance and trans-QTP speed (calculated as the trans-QTP distance/trans-QTP duration). For this study, the QTP was defined as the areas above 3,000 m in altitude, and we contoured it in ArcView GIS and computed the trans-QTP distance of each migration route using the XTools extension. To assess the topographical preference for migration, we determined the mean altitude of each route, which was automatically generated in Google Earth by plotting the migration routes in the software. The southern and the northern QTP is crossed by two representative latitudes, 29°N and 35°N, so we recorded the longitudes crossing these two latitudes to test the spatial flexibility of the repeated migration routes across the QTP.

Longitudinal detours in migration routes

To assess the degree of longitudinal detours in migration routes by individuals within each season, we compared each observed route with 1,000 randomized routes that were automatically generated by rearranging each segment of the observed route using the Alternate Animal Movement Route extension for ArcView GIS (Jenness, 2005; López-López, García-Ripollés & Urios, 2014). Each randomized route had the same number and sequence of segments but a different segment orientation from the observed route. We recorded the longitudes crossing each 3° latitude interval from 23°N to 44°N for each observed route and the 1,000 simulated routes (i.e., the randomly generated routes). If a bird doubled back during the migration, resulting in two or more longitude values crossing a specific latitude, the most longitudinally deviated value was used. If the observed longitudes for a bird fell below the 2.5th percentile or above the 97.5th percentile (equivalent to a two-tailed test with P = 0.05) of the 1,000 randomly generated longitudes, the observed longitudes were considered to have significantly deviated from the possible routes (López-López, García-Ripollés & Urios, 2014) (Fig. S1).

To measure the extent of longitudinal detours, we calculated the linear distance of the maximum longitudinal deviation of each observed route from the expected shortest route at 3° latitude intervals from 23°N to 44°N. When the gulls overshot the breeding site or wintering site, we calculated the linear distance of the maximum longitudinal deviation of the overshot segment from the breeding site or wintering site. In this case, we also calculated the linear distance of the maximum latitudinal deviation of overshot stopover sites from the breeding site or wintering site as a measurement of latitudinal detour (Fig. S2).

Detours and habitat association

Pallas’s Gulls exclusively use open water as stopover sites for refuelling, so the availability of open water may have profound influence on migration route selection (Zhang et al., 2011a). We analysed the proximity to water bodies based on the China National River System Data Set (http://www.pudn.com/downloads608/sourcecode/graph/detail2478372.html), which comprises information on third-order rivers and lakes with areas ≥0.01 km2 in China. The composition of lakes and rivers was extracted from zones with 100 km buffers around the observed and expected shortest routes in ArcView GIS, and the differences between the zones were then compared to assess migration detours and habitat associations. We did not analyse detours and habitat associations for the southern stretches of the route outside China because all individuals were funnelled through a narrow corridor to the wintering site, which was spatially near the expected shortest routes.

Statistical analyses

We checked the data for normality to determine whether the use of parametric or nonparametric tests. We used paired t-tests or Wilcoxon signed-rank tests to compare the differences between average migration speed and trans-QTP speed for each route and determine the differences in proximity to water bodies between the observed and expected shortest routes. We compared the longitudinal variation across the QTP by ANOVA, with individual and season as factors, and we analysed the seasonal differences in migration route and timing parameters using GLM mixed models, with individual as a random factor. We also used GLM mixed models to compare the between- and within-individual variation in the timing and route of repeated migration tracks. Based on the derived variance components, we calculated the repeatability (i.e., intra-individual correlation coefficient; Lessells & Boag, 1987) of migration timing and route to measure how consistently individuals differed from each other. All statistical analyses were performed using the SPSS statistical software package (version 22.0, IBM 2013), and the results were calculated as a mean ± SD, with a 0.05 level of significance based on two-tailed tests.

Ethical note

All data collected as part of this study were approved by the National Bird Banding Center of China (No. NBBC20060407). The field work was approved by the State Forestry Administration (No. 33 Forestry Protection (2002)).

Results

The complete data set for the four tracked Pallas’s Gulls contained 2196 positions, consisting of 32.5% high-quality locations (LC 1–3). Overall, we obtained data for 22 complete migratory journeys (12 in autumn and 10 in spring), with the four birds repeatedly tracked across the QTP between Qinghai Lake in north-western China and Bangladesh Bay for two to four consecutive years (Table 1, Fig. 1).

Table 1 Description of parameters in migration timing and route of four Pallas’s Gulls across Qinghai-Tibetan Plateau determined by satellite telemetry during 2006–2009.

	Parameters	G1	G2	G3	G4	All birds	
Autumn	No. journeys	4	3	2	3	12	
Departure date	10 Aug ± 5	12 Aug ± 12	13 Aug ± 5	4 Aug ± 1	9 Aug ± 7	
Arrival date	7 Nov ± 4	11 Nov ± 6	30 Nov ± 4	23 Nov ± 1	16 Nov ± 10	
Duration	89 ± 9	91 ± 8	110 ± 9	111 ± 1	99 ± 12	
Trans-QTP duration	87 ± 9	40 ± 44	33 ± 4	26 ± 2	51 ± 33	
Shortest distance	1,815 ± 7	1,895 ± 27	1,906 ± 131	2,116 ± 84	1,925 ± 133	
Cumulative distance	2,009 ± 107	3,375 ± 1,161	3,333 ± 1,010	4,265 ± 170	3,135 ± 1,086	
Trans-QTP distance	1,304 ± 246	1,649 ± 211	1,245 ± 79	1,913 ± 175	1,533 ± 329	
Speed	23 ± 2	38 ± 14	30 ± 7	38 ± 2	32 ± 10	
Trans-QTP speed	15 ± 3	130 ± 154	38 ± 7	75 ± 10	63 ± 81	
Straightness	0.91 ± 0.05	0.62 ± 0.26	0.61 ± 0.22	0.50 ± 0.02	0.68 ± 0.22	
Mean altitude	3,051 ± 350	2,693 ± 276	2,257 ± 20	2,498 ± 11	2,691 ± 372	
Spring	No. journeys	3	3	2	2	10	
Departure date	2 Mar ± 8	19 Mar ± 2	28 Mar ± 3	19 Mar ± 2	15 Mar ± 11	
Arrival date	4 Apr ± 3	11 May ± 14	14 May ± 22	26 Mar ± 4	21 Apr ± 24	
Duration	34 ± 8	53 ± 16	47 ± 19	7 ± 1	37 ± 20	
Trans-QTP duration	7 ± 4	50 ± 18	13 ± 6	6 ± 1	21 ± 22	
Shortest distance	1,831 ± 31	1,856 ± 19	2,008 ± 216	2,256 ± 1	1,959 ± 186	
Cumulative distance	3,177 ± 191	2,199 ± 125	3,618 ± 1	2,415 ± 136	2,819 ± 602	
Trans-QTP distance	1,229 ± 45	1,605 ± 142	2,279 ± 572	1,458 ± 207	1,597 ± 446	
Speed	98 ± 26	44 ± 13	85 ± 35	351 ± 52	130 ± 121	
Trans-QTP speed	198 ± 85	35 ± 13	196 ± 54	246 ± 23	158 ± 98	
Straightness	0.58 ± 0.03	0.85 ± 0.05	0.55 ± 0.06	0.94 ± 0.05	0.73 ± 0.17	
Mean altitude	2,224 ± 108	3,069 ± 131	3,111 ± 462	2,543 ± 264	2,719 ± 449	

Figure 1 Map showing the repeat migration tracks of four Pallas’s Gulls between Qinghai Lake, China, and Bangladesh Bay determined by satellite telemetry during 2006–2009.

Solid dark grey lines indicate spring migration routes and dashed dark grey lines indicate autumn migration routes. Background grey shows the Qinghai-Tibetan Plateau above 3,000 m in altitude. A total of 12 complete autumn and 10 spring migration routes were recorded. (A–D) represent for G1–G4, respectively.

Spatial patterns of detours

Notable detours were observed along the migration routes in both seasons (Fig. 1). Gulls exhibited increasing longitudinal detour extents from south to north across the migration routes in both autumn (chi-square test for trend, X2 = 6.627, df = 1, P = 0.010) and spring (X2 = 4.151, df = 1, P = 0.042) (Fig. 2), with the largest detours being 1,068 km at 38°N in autumn and 685 km at 32°N in spring. Similarly, the percentage of the observed routes that significantly deviated from the simulated routes increased from south to north in autumn (X2 = 5.955, df = 1, P = 0.015).

Figure 2 Bar plot showing the mean longitudinal deviation of observed routes from the expected shortest routes in latitudinal bins of three degrees, for autumn (black) and spring (white) migrations.

Error bars show standard deviation, pie charts show proportion of routes that deviated significantly (p < 0.05) from 1,000 randomised routes starting and ending at the same point.

After departure from Qinghai Lake in seven (n = 12) autumn migration seasons, three of the four gulls exhibited notable northward migration outside of the QTP to stopover sites at lower altitudes (Δ altitude = 2,337 ± 385 m compared with Qinghai Lake, n = 7; Fig. 1), with an average latitudinal detour of 627 ± 275 km (range = 133–799 km, n = 7). Similarly, in five (n = 10) spring migration seasons, two of the four gulls overshot the breeding grounds to farther northern stopover sites at lower altitudes (Δ altitude = 2,029 ± 139 m compared with Qinghai Lake, n = 5) and then reverse migrated back to Qinghai Lake, with an average latitudinal detour of 168 ± 24 km (range = 140–193 km, n = 5).

Relative to the expected shortest routes, gulls travelled 1,210 ± 1,002 km (range = 81–2,240 km, n = 12) or 61.3 ± 50.4% (range = 4.5%–123.2%, n = 12) of additional detour distance in autumn and 860 ± 648 km (range = 63–1,762 km, n = 10) or 45.3 ± 34.6% (range = 2.8%–95.0%, n = 10) in spring. Gulls covered significantly longer trans-QTP distances along the observed routes than along the shortest routes both in autumn (paired t-test, t = 6.52, df = 11, P < 0.001) and in spring (t = 3.97, df =9, P = 0.003).

Temporal patterns of detours

Gulls used different routes in different seasons and followed a loop migration pattern (Fig. 1). On average, gulls followed larger longitudinal detours in autumn than in spring at all eight measured latitudes from 23°N to 44°N (Wilcoxon signed-rank test, Z =  − 2.521, P = 0.008) (Fig. 2). The effect of season on longitudinal variation was strong at 35°N (ANOVA, F = 16.46, df = 1, P = 0.001, h2 = 0.540) but weak at 29°N (F = 0.68, df = 1, P = 0.424), reflecting that the loop migration primarily occurred across the northern part of the QTP. A significant individual × season interaction was detected both at 35°N (F = 9.47, df = 3, P = 0.001, h2 = 0.670) and 29°N (F = 18.33, df = 3, P < 0.001, h2 = 0.797), indicating that different individuals migrated in different loops or even followed loops in different directions (i.e., clockwise or counter clockwise; Fig. 1).

No significant seasonal differences were observed in the migration route parameters, including cumulative distances, trans-QTP distances, straightness and mean altitude (GLM mixed models, F1,20 = 0.10–0.31, P = 0.618–0.778). However, the spring migration duration was significantly shorter than that in autumn (GLM mixed models, F1,20 = 22.38 P = 0.018). Additionally, no significant differences were detected in trans-QTP duration, average speed and trans-QTP speed (GLM mixed models, F1,20 = 0.23–2.88, P = 0.188–0.424) between autumn and spring (Table 1).

Gulls migrated faster across the QTP than during the entire journey both in autumn and spring (Table 1), but the differences were not significant (Wilcoxon signed-rank test, autumn: Z = 1.413, P = 0.170; spring: Z = 0.867, P = 0.418).

Detours and habitat associations

During the autumn migration, we found significantly increased (paired t-test, t = 2.90, df = 11, P = 0.015; Fig. 3) proximity to water bodies along the observed detour routes (1.3 ± 0.7%, n = 12) than on the expected shortest routes (0.7 ± 0.2%, n = 12). However, we did not detect a significant difference in proximity to water bodies (paired t-test, t = 0.49, df = 9, P = 0.635) between the observed detour routes (0.8 ± 0.7%, n = 10) and the expected shortest routes (0.7 ± 0.1%, n = 10) in spring.

Figure 3 Line plots showing comparison of proximity to water bodies along expected shortest routes and observed routes during autumn (A) and spring (B) migration of Pallas’s Gulls.

* P < 0.05, n.s. indicates P ≥ 0.05. Significance levels were based on paired t-test.

Individual consistency in detours

The effect of the individual was significant for almost all migration variables, except for several timing variables in autumn (Table 2). Tests of within-individual variation showed that migration variables were more repeatable in spring than in autumn and in route than in timing. In autumn, the departure date was highly flexible (r =  − 0.07), but the arrival date was highly repeatable (r = 0.85); however, both the departure and arrival date were consistent, for which individuals differed between years by an average of ± 6.1 d and ± 3.5 d, respectively. In spring, both the departure date (r = 0.84) and arrival date (r = 0.78) were highly repeatable; however, gulls showed consistent departure date and somewhat flexible arrival date, for which individuals differed between years by an average of ± 4.5 d and ± 14.0 d, respectively. The longitude crossing 35°N (rautumn = 0.82, rspring = 0.91) and 29°N (rautumn = 0.75, rspring = 0.89) in both seasons had high repeatability. In autumn, the within-individual variation of the longitude crossing 35°N and 29°N between years averaged 201 km (6.4% of the accumulated distance) and 224 km (7.1% of the accumulated distance), respectively. In spring, the variation averaged 84 km (3.0% of the accumulated distance) and 138 km (4.9% of the accumulated distance), respectively. Therefore, gulls showed higher consistency in routes across the QTP in spring than in autumn.

Table 2 GLM mixed model testing the effect of “individual” on migration timing and route with repeatability (r) values.

Season	Parameters	df	F	r	P	
Autumn	Departure date	3, 8	0.80	−0.07	0.527	
Arrival date	3, 8	17.35	0.85	0.001**	
Duration	3, 8	7.06	0.67	0.012*	
Trans-QTP duration	3, 8	5.28	0.59	0.027*	
Speed	3, 8	3.39	0.45	0.074	
Trans-QTP speed	3, 8	1.35	0.11	0.324	
Cumulative distance	3, 8	6.42	0.65	0.016*	
Trans-QTP distance	3, 8	6.72	0.66	0.014*	
Straightness	3, 8	4.48	0.54	0.040*	
Mean altitude	3, 8	5.16	0.59	0.028*	
Longitude crossing 29°N	3, 8	9.74	0.75	0.005**	
Longitude crossing 35°N	3, 8	14.55	0.82	0.001**	
Spring	Departure date	3, 6	13.76	0.84	0.004**	
Arrival date	3, 6	9.73	0.78	0.010*	
Duration	3, 6	5.57	0.65	0.036*	
Trans-QTP duration	3, 6	10.05	0.79	0.009**	
Speed	3, 6	45.18	0.95	0.000***	
Trans-QTP speed	3, 6	7.53	0.70	0.019*	
Cumulative distance	3, 6	51.02	0.95	0.000***	
Trans-QTP distance	3, 6	6.64	0.73	0.025*	
Straightness	3, 6	35.78	0.94	0.000***	
Mean altitude	3, 6	8.62	0.76	0.014*	
Longitude crossing 29°N	3, 6	20.06	0.89	0.002**	
Longitude crossing 35°N	3, 6	27.00	0.91	0.001**	
Notes.

* P < 0.05.

** P < 0.01.

*** P < 0.001.

Discussion

We reported on the detours of a long-distance migrant confronted with the geographical barrier of the QTP. The maximum distance of the observed route was more than double that of the expected shortest route. Unusually, gulls followed notable latitudinal detours that accounted for up to half of the total latitudinal extent between the breeding and wintering grounds, and to the best of our knowledge, this detour is one of the largest in avian migration (Alerstam, 2001; Mellone et al., 2011). Although our dataset was relatively small, this study provides important insights into the cost of migratory detours in terms of the annual energy budget of an individual as well as the factors underlying these detours across a high plateau.

Migratory birds are expected to detour to shortcut across part of a barrier because non-stop, long-distance flights demand extra energetic costs to transport the required heavy fuel loads (Alerstam, 2001). However, although detours occurred, the Pallas’s Gulls extended rather than reduced the trans-QTP distance compared with the shortest route because the birds could refuel at a couple of stopover sites on the QTP (Zhang et al., 2014). How did gulls compensate for the extra energetic costs of the detours, and what was the justification for the latitudinal detours to low-altitude stopover sites, as climbing up and down the QTP might have incurred extra costs?

Figure 4 The movement locations of Pallas’s Gulls and habitats at detoured stopover sites with low altitude indicated by Google earth.

(A) Qinggeda Lake in Wujiaqu City, (B) Lakes in Wulumuqi City, (C) Hongyashan Reservoir in Minqin County, (D) Yellow River in Yinchuan City. All these stopover sites are characterized by water bodies which were used as key fuelling sites for moulting and trans-QTP flights.

Birds sometimes follow detours or even perform reverse migrations to seek better fuelling sites before undertaking long-distance migratory flights across ecological barriers (Lindström et al., 2011; Smolinsky et al., 2013; D’Amico et al., 2014). In our study, gulls (G2, G3 and G4) migrated northward to stopover sites at significantly lower altitudes in autumn and stayed for considerable durations (range = 50–83 days; Data S1). The energetic costs of locomotion are reduced at lower altitudes because a lower metabolic rate can be maintained (Scott, 2011; Hawkes et al., 2013). All these stopover sites were identified as water bodies, including lakes (Figs. 4A–4B), reservoir (Fig. 4C) and stretches of the Yellow River (Fig. 4D). We suggest that gulls might use the different food compositions at low-altitude stopover sites to their advantage, changing their diet to rapidly increase body mass before migrating across the challenging plateau (Bairkein, 1998; Mcwilliams et al., 2004). Additionally, these areas might also be used for moulting since gulls stayed there from August to October and many waterbirds tend to follow detour in moult migration (Mosbech et al., 2012; Solovyeva et al., 2014). Considering that Pallas’s Gulls made an average of two stopovers during a 3,000-km journey (Data S1), i.e., employed a “skipping” migration strategy (Piersma, 1987), the quality of stopover sites is especially crucial for long-distance, non-stop flights, so the search for better fuelling sites is probably the drivers underlying the detours. The results of our GIS-based habitat analysis, which revealed increased proximity to water bodies along autumn migration routes relative to the shortest routes, also supported this conclusion. Along spring migration routes, however, a weak detour-habitat association was detected. This means that habitat availability might be one, but not the only driver shaping detour extent (Hahn et al., 2014), and migration routes were shaped under different mechanisms during autumn and spring.

Weather conditions are another important factor shaping migration detours. To avoid the influence of cold weather, Relict Gulls (Larus relictus) follow more southward routes in spring compared to autumn to seek favourable fuelling sites (Liu et al., 2017), and in case of an unexpected adverse weather event, birds might even perform reverse migration (Senner et al., 2015). In our study, the onset of spring migration for G1 was nearly 20 days earlier than that of the other three individuals (Table 1), so consequently, G1 overshot its frozen breeding ground at Qinghai Lake and stayed at a warmer north-eastern stopover site on a stretch of the Yellow River from mid-March to early April during 2007–2009 (Fig. 5).

Figure 5 Line plots showing differences in ground surface temperatures between the breeding ground at Qinghai Lake (solid lines) and overshot stopover site on a stretch of the Yellow River stretch (dashed lines).

This figure exemplifies the influence of cold weather on detours in spring. One gull (G1) overshot the frozen breeding ground at Qinghai Lake and stayed at the warmer Yellow River from mid-March to early April in 2007 (red lines), 2008 (blue lines) and 2009 (green lines). Temperature were extracted from the local meteorological stations from NOAA (available at https://gis.ncdc.noaa.gov/maps/ncei#app=clim&cfg=cdo&theme=hourly&layers=1&node=gis).

The Pallas’s Gulls followed different detours between seasons and therefore exhibited loop migration, which has also been documented in other Larus species (Liu et al., 2017). Avian migrants often face seasonal differences in the prevailing winds along their migration flyway. To achieve optimal migration, bird might overcompensate and overdrift for side winds and consequently follow detours and seasonally different routes (Vansteelant et al., 2017). In certain geographical areas, assistance (or hindrance) by the wind may be persistent during migratory flight, depending on the migratory direction of the birds relative to the prevailing wind conditions. It has been documented that supporting winds are significantly more frequent during spring migration than in autumn and up to twice as frequent at higher altitudes, which substantially impacts the timing of seasonal migration in many birds (Sinelschikova et al., 2007; Kemp et al., 2010). Although similar distances were covered, the Pallas’s Gulls in our study migrated significantly faster in spring than in autumn; at their extremes, the gulls accomplished the spring migration in less than six days at a migration speed of more than 360 km/day. This strategy is crucial for gulls that communally breed on islands because nest site competition profoundly affects reproductive success; e.g., nests of late breeders suffer flooding at the foots of islands (Burger & Shisler, 1980; Miao, 2014).

Our results showed significant between-individual variation but high within-individual repeatability in migration timing and route in both seasons. Although the high repeatability of spring arrival date mispresented the consistency due to the relatively high between-individual variation (i.e., population variation; Conklin, Battley & Potter, 2013), the departure date and arrival date were in general quite consistent between years. This indicated a stronger influence of endogenous schedules over local environmental conditions, which was consistent with the results of many previous studies (Vardanis et al., 2011; Stanley et al., 2012; López-López, García-Ripollés & Urios, 2014). Compared with migration timing, Pallas’s Gulls were even more consistent in their migration route, although different individuals used different detour schemes. Birds could benefit from repeated use of familiar routes and stopover sites by reducing predation risk (Yoder, Marschall & Swanson, 2004) and energetic costs in a search for new sites, which may be of particular significance under the challenging environmental conditions of the QTP.

In conclusion, Pallas’s Gulls confronted with the QTP exhibited notable detours that differed significantly in extent between seasons. Despite the extra energetic costs of repeatedly climbing up and down the QTP, gulls exhibited latitudinal detours to significantly lower-altitude stopover sites for considerable fuel loading, which has not been observed in other typical species, such as Bar-headed Goose Anser indicus and Black-necked Crane Grus nigricollis that migrate across the QTP (Zhang et al., 2011b; Liu et al., 2012). Further investigation is needed to shed light on this novel migration strategy under the challenging environment of the QTP.

Supplemental Information

Data S1 Migration details of four Pallas’s Gulls across Qinghai-Tibetan Plateau during 2006–2009

Major stopover sites are showed in color.

Click here for additional data file.

Figure S1 An example of analysis of simulated route

Spatial deviation in migration routes was analyzed by comparing observed route (red line) with 1,000 simulated routes (blue lines) obtained by randomly distributing the real route segments (see details in ‘Methods’). The number in the right column shows the percentile at which the longitude value of the observed route distributes in the 1000 longitude values of the simulated routes at each 3° latitude interval from 23°N to 44°N. If the percentile ia above or below 2.5% and 97.5%, the observed route was considered to deviate significantly from the simulated route at the specific latitude.

Click here for additional data file.

Figure S2 Measurement of the longitudinal and latitudinal detour extent

Click here for additional data file.

We thank the Qinghai Lake National Nature Reserve Bureau for logistically support of the fieldwork. We thank Yunqiu Hou, Ming Dai, Kai Shan, Yuansheng Hou and Yanming Wang for their assistance with the fieldwork and Guang Deng for his assistance with the GIS analysis. We also thank Krysta Black-Mazumdar and Sarah Conte for improving the English as well as Lucy Hawkes and an anonymous reviewer for their valuable and constructive comments on earlier drafts of the manuscript.

Additional Information and Declarations

Competing Interests

Author Contributions

Animal Ethics

Field Study Permissions

Data Availability

The authors declare there are no competing interests.

Dongping Liu conceived and designed the experiments, performed the experiments, analyzed the data, wrote the paper, prepared figures and/or tables, reviewed drafts of the paper.

Guogang Zhang conceived and designed the experiments, performed the experiments, analyzed the data, reviewed drafts of the paper.

Hongxing Jiang conceived and designed the experiments, performed the experiments, reviewed drafts of the paper.

Jun Lu conceived and designed the experiments, contributed reagents/materials/analysis tools, reviewed drafts of the paper.

The following information was supplied relating to ethical approvals (i.e., approving body and any reference numbers):

All data collected as part of this study were approved by National Bird Banding Center of China.

The following information was supplied relating to field study approvals (i.e., approving body and any reference numbers):

The field work was approved by the State Forestry Administration (No. 33 Forestry Protection (2002)).

The following information was supplied regarding data availability:

The raw data has been provided as a Supplemental File.

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
