# Peer review of "Detours in long-distance migration across the Qinghai-Tibetan Plateau: individual consistency and habitat associations"

_PeerJ, doi:10.7717/peerj.4304_

## Round 0.1 · original submission · Major Revisions

Both reviewers and I agree that this is an interesting paper that addresses an important topic.

As you will see, both reviewers have provided thoughtful and detailed reviews, and raised several issues that need to be addressed. In particular I agree with reviewer 2 that you should be using great circle distance for your migration calculations, given the scale of movements involved. Reviewer 1 also notes several areas where clarity could be improved and where terminology should be adjusted.

From my own reading of the paper, it struck me as sub-optimal that your statistical analyses and quantification of among and within-individual variance components did not use GLMMs with individual as a random effect. You will note that Reviewer 1 has raised the same issue, and it needs to be given some treatment. This approach would allow you to test predictors of interest and estimate variance components in a single framework, rather than conducting multiple separate tests.

Finally, I agree that the use of the word 'extreme' may not be entirely appropriate as an umbrella term for the migration of these gulls, as i) the total detour relative to the shortest distance is only ~2x, and ii) there is a lot of among individual variation in magnitude of detour. Please adjust the use of this term accordingly, as suggested by Rev. 1.

·

Basic reporting

No comment in addition to review below

Experimental design

No comment in addition to review below

Validity of the findings

No comment in addition to review below

Additional comments

Major comments:
The manuscript presents some exciting data showing how Pallas gulls may move around a major ecological barrier during their annual migrations, and present the surprising finding that the gulls move up to twice as far as necessary to reach their wintering grounds, making sideways detours of considerable distance. The data appear to have been analysed appropriately (but see my comments about pseudo-replication, which need dealing with) and the results are of interest to the wider community. The basic reporting is good, the experimental design is fine and the validity of the findings are adequate, but see below for some thoughts that I think need addressing:

• I am not sure that “extreme detours” is accurate. Yes, the birds are making a detour, but it’s double the distance, not 10 times, which might be more clearly “extreme”. Granted, it is difficult to set a cut-off, but perhaps this aspect just needs to be moderated by deleting “extreme” from the title and text.
• “Habitat availability” (e.g. abstract line 16, methods line 144, etc). The proximity to water bodies that may or may not be suitable for gulls to use during migration is not the same thing as habitat suitability. Indeed, there is a whole Niche modelling field that uses that term and that is therefore implied here, when you have made no such analysis. I would be honest here and call it what it is – proximity to water bodies. You don’t know that these water bodies are suitable (they could be occupied by people (line 299), polluted, dried up etc). so just keep it simple.
• Reverse migration – I am not sure what you mean by this? You give a definition on line 131 but in bird migration it has another meaning. I would suggest instead saying something like “in case the bird doubled back on itself during the migration”
• As I see it, springtime, northward migrating gulls don’t exhibit much deviation from the shortest route. This seems sensible if the gulls were attempting to get to optimal breeding sites before competitors each year. This also could explain why the arrival date is consistent too (line 231). Autumn, southward migrating gulls, however, seem to be the ones making the detours. These can be considerable and one wonders what they might be doing there. It would seem sensible to use Google Earth to at least describe the land that they are deviating to, is it agricultural? Water bodies? You essentially hint at this on line 261 and 279 but I think it could and should be more explicitly discussed in a dedicated paragraph. You introduce several reasons for detours nicely in the introduction, I would expect to have seen each hypothesis individually discussed.
• Navigation – This comes in at the end of the discussion and I am not sure that it is warranted. The study is sufficient without including discussion of putative navigation strategies. Certainly there are not robust data with which to test this. I would suggest removing it from the manuscript entirely.

Minor comments:
Abstract:
Line 8 – change “the” to ‘their’
Line 13 – I think it would be more honest here to say that you tracked four gulls, but for repeated years. The 22 migration seasons instead makes it sound like there is more data than there is.
Line 16 – see comment above about habitat availability.
Line 26 – see comment above about navigation.

Introduction:
Line 45 – “detoured with a short cut” is an oxymoron – can you rewrite this to whatever you mean?
Line 49 – also insert reference to Hawkes et al. (2012) Proc Roy Soc B 108, 9516 who showed that bar-headed geese detour considerably from the shortest route to avoid strong winds.
Line 54 – change “on extreme plateau” to ‘at high altitude’. Likewise for line 57.
Line 75 – needs a reference for this statement.
Line 77 – you should also cite Pennycuick (2008) Modelling the Flying Bird, Academic Press.

Methods:
Line 98 – what does “complete” mean?
Line 129 – need to very clearly describe these as randomly generated longitudes and observed bird longitudes throughout.
Line 132 – This is a bit long winded at present. How about instead: “If the observed bird longitudes fell below the 2.5th percentile, or above the 97.5th percentile of the 1,000 randomly generated longitudes, the observed bird longitude was considered to significantly deviate from the migratory route that might have been expected.”
Line 137 – the linear distance between
Line 139 to 143 – this isn’t clear to me, I don’t understand what other metric you are collecting here and how it differs from line 137?
Line 144 – see comment above about ‘habitat suitability’
Line 146 – needs a reference for this point
Line 155 - In order to compare between northward (spring) and southward (autumn) migrations statistically as you outlined here, you need to account for individuals (as you show on line 228). Line 81 states “from two to four years” suggesting that three of the four birds may have contributed more data to the analyses and plots than the one that only lasted two years. Essentially, data presented in Figure 3, for example, may be pseudo-replicated. I would suggest something more sophisticated than a t-test should be used, at least something like a GLM where you can include individual as a random factor. For plotting, you should ensure that the same number of trips per animal go in, i.e. if one gull was only tracked for two trips, only two from the other gulls should go into the plot as well and the other trips excluded.

Results:
Line 175 – Give the n per individual (with only four birds seems little point in giving a mean +/- s.d.). Can you also give the duration over which the birds were each tracked as well?
Line 178 – is “two-four” a typo?
Lines 183, 184, 186, 187, 190, – there is a statement on each of these lines (e.g. “larger”) that needs to be supported by a statistical result. From this point onwards, the stats have been reported well.
To me, line 184 is the major finding of the whole study and needs to be pulled out to be more prominent, e.g. by start the paragraph at line 205 with it, and calling that paragraph “Temporal patterns of detours” so it marries well with the previous paragraph heading.
Line 189 – add (n=x per gull). The 75% here is unnecessary, just say three of the four gulls (same for line 193).
Line 198 – how many per individual? I am hoping it is three each!
Line 200 – I would delete “In contrast…barriers after detours” and start the sentence at “Gulls covered”
Line 205 – then you need to give the effect size here as well, it would be in km.
Line 206 – a loop migration is something different to what you have observed.

Discussion:
Line 249 – this sentence is muddled – you can’t detour with a shortcut? I suggest rewriting it to whatever you mean.
Line 251 – likewise muddled
Line 253 – what does “inhospitable at ground” mean?
Line 254 – I would argue that the availability of stop-over sites doesn’t necessarily mitigate the altitudinal climb required for the gulls to get onto the QTP from Bangladesh.
Line 263 – some typos here, I think you mean “might not be the only...”
Line 266 – do you mean significantly lower altitudes?
Line 269 – but are gulls a species that undertakes considerable fuel loading prior to migration? If they stop lots of times they may not need to.
Line 270 – this should be rewritten as something more like: “Additionally, the energetic costs of locomotion are reduced at lower altitudes”
Line 302 – this sentence is a little confusing (“trans-barrier” comes up for the first time) – what do you actually mean here? Perhaps that one might imagine that once a gull had climbed onto the QTP it wouldn’t fly back off it again any sooner than it absolutely had to? I suggest changing this sentence.
Line 304 – I disagree and think the most interesting is the extent of the detour in northwards versus southwards migration.

Figures and Tables
Figure 1 – does this need to be in colour? Given that much of the world prints to black and white printers, it would be better to produce the figure in black and white if possible. The land could have a black outline and needs some country labels. The tracks could be solid dark grey for northwards and dashed dark grey for southwards migrations (the 3,000 metre polygon would still be visible and clearly distinguishable). I think you should plot the shortest route on the map as well so we can see the differences – this could be a thick black line. I would also reduce the number of tick marks and labels to just 4-6 on each axis. The legend should also start “Map showing…”

Figure 2 – I think the legend could more clearly explain the plot, it’s not clear enough right now and doesn’t mention the randomisation routines. I believe something like this might be more helpful: “Bar plot showing the mean longitudinal deviation from the shortest route in latitudinal bins of three degrees, for northward (blue) and southward (red) migrations. Error bars show standard deviation, pie charts show proportion of routes that deviated significantly from 1000 randomised routes starting and ending at the same point, as measured using a XX test and p<0.05.” Again, this plot could be in black and white instead of colour.

Figure 3 – I would shade the boxes in grey, which makes the plot easier to read. I would also consider adding notches to denote significant differences. I am not sure what “Turkey style” whiskers are? Given the sample size, I am surprised the data met a normality test to use a t-test and there is no mention of a normality test in the methods. Can you check they are normally distributed and if not, use a Mann-Whitney? The legend should also start “Boxplots showing…”. It is also not clear how pseudo-replicated the boxplots might be, since I imagine there is more than one trip per gull in there (otherwise there are only four datapoints making each box). You should use the same number of trips per gull and explicitly state this in the legend.

Figure 4 – See my earlier comment. Calling this “suitable habitat” may be misleading, or rather, it would be more accurate just to say “proximity to water bodies” as that is what you have actually measured. Again, this plot could be in black and white instead of colour. The legend should also start “Line plots showing…”

Figure 5 – I feel like there should be better way to show these data, or perhaps just to mention it in the text instead. If you do keep the figure, I would change the legend to “satellite derived ground surface temperatures”, which I believe is what you actually used, although I don’t think there is any mention of where these data came from in the methods? I am also not sure that “overshot” is accurate (did they overshoot or did they mean to go there? Not possible to know). The legend should also start “Line plots showing…”.

Reviewer 2 ·

Basic reporting

The English could be improved throughout. A few citations could be changed/added as well. (See General Comments.)

Experimental design

I believe that the migratory detours need to be calculated in relation to a great circle route as opposed to a straight line route. (See General Comments.)

Validity of the findings

I have some minor quibbles related to the Discussion section. (See General Comments.)

Additional comments

General Comments
I like this paper a lot. I think that it details an interesting dataset and offers some novel insights into the drivers of migratory route choice in long-distance migratory birds. I have only one ‘major’ concern: I think the migratory deviations should be calculated in relation to the great circle distance between non-breeding and breeding sites, as opposed to the straight distance. Otherwise, my only other general suggestion is that the writing needs to be tightened up throughout the manuscript and especially at the spots I note below. All in all, well done!

Specific Comments
Lines 13-14: I do not fully understand the second clause of this sentence.

Lines 45-47: I do not entirely understand this sentence.

Lines 66-70: I am not sure if this large vs. medium-sized bird dichotomy really exists any more in terms of tracking studies. Tons of tracking studies of all sorts have focused on ‘less-than-large species’ recently, including a number from the shorebird literature related to within and between individual repeatability (e.g., Conklin et al. 2013, PLoS One). I think that you can probably delete these two sentences if you would like.

Line 77: See also Parr et al. 2017 in Journal of Avian Biology for another recent trans-Himalayan migration example.

Line 89: I presume that the gulls were breeding at Qinghai Lake?

Line 107: What were the duty cycles on the PTTs?

Lines 111-112: Shouldn’t you calculate the great circle distance instead of a straightness index? Birds have frequently been shown to adhere to great circle routes.

Lines 118-119: Do you present these data in the results?

Lines 145-146: Citation?

Lines 243-246: See comments below about moult migration.

Lines 247-248: I am unsure of how your study provides any information on flight range capacity. Given that your gulls were stopping over along their route, these detours were not part of non-stop flights and thus were not extending our knowledge of truly long-distance flights. What your data do shed light on is how expensive migratory detours actually are in terms of the annual energy budget of an individual, but that is not their flight range capacity, per se.

Lines 266-270: I think that you need to mention the literature on moult migration here. See, for instance, Mosbech et al. 2012 in Polar Biology for a discussion of this behaviour in Little Auks (Alle alle). Even if this is not a plausible explanation for the northward fall movements that you observed in Pallas’ Gulls, numerous waterfowl and seabirds exhibit large detours as part of their moult migrations and these should at least be noted.

Lines 272-274: Nilsson and Sjöberg 2015 were generally unable to correlate observed reverse migrations over Falsterbo with any weather conditions. I think that Senner et al. 2015 in Journal of Animal Ecology is a stronger citation here.

Lines 278-279: I do not think that faster springer migrations necessarily mean that a species is a time-minimizer. For one thing, wind regimes could simply be more favorable in spring for migration, enabling individuals to migrate faster than in fall, but saying nothing about the relative use of time- vs. energy-minimization strategies.

Lines 281-283: Citation?

---

## Round 0.2 · Minor Revisions

Both the reviewer and I agree that this draft is greatly improved over the original version.

Having been assessed by one of the original reviewers, there are only a few minor issues that need to be cleared up. Please pay specific attention to the issue of repeatability vs consistency identified by the reviewer.

Reviewer 2 ·

Basic reporting

See general comments below.

Experimental design

See below

Validity of the findings

See below

Additional comments

General Comments:

The authors adequately addressed all of my concerns from the first draft of the manuscript — well done! However, I noticed one other statistical issue — regarding the relationship between repeatability and consistency — that I missed in the first draft (my apologies) and which I think needs to be addressed. Otherwise, I just have suggestions about ‘word-smithing.’

Specific Comments:

Abstract: I think that the Abstract could be shortened significantly (e.g., Lines 25-29 could be deleted).

Line 9: I think that the use of the word ‘adaptive’ here is odd.

Lines 10-12: This is a bit of a run-on sentence.

Lines 28-30: Do you have evidence to support the first part of this sentence? If not, you should use the word ‘may’ instead of ‘can.’

Line 39: Should be ‘energetic’ instead of ‘energy’

Lines 41-43: This is an awkward sentence.

Line 44: What is a ‘notably’ heterogeneous environment? This sentence, in general, is unintelligible to me.

Lines 51-53: I think that you want to cite Gill et al. 2009 (Proceedings of the Royal Society B) here as a counterpoint, since large-fuel loads can enable birds to avoid stopping in dangerous areas, etc.

Line 64: Yes, low PPO2 is omnipresent over high plateaus, but plateaus present other barriers as well (depending on the species) and may offer fewer stopover areas than other regions.

Line 66: You are referring to phenotypic flexibility here, not phenotypic plasticity (sensu Piersma and Drent 2003, Trends in Ecology and Evolution).

Lines 258-269: I apologize for not catching this in the first draft of the paper, but there is an important distinction between ‘consistent’ and ‘repeatable’ in statistical terms. Repeatability — as a statistical measure — is always comparing an individual’s behavior to the expected amount of variation exhibited by the entire population. This means that low amounts of among-individual variation can lead to left-skewed measures of repeatability, while high amounts of among-individual variation can lead to right-skewed measures of repeatability. (See Conklin et al. 2013 (PLoS One) for an extensive discussion of this issue.) In this paper, there are two potential issues: (1) Four individuals do not necessarily provide a representative sample of the amount of population-level variation existing among Pallas’ Gulls, and thus your measures of repeatability may be right-skewed. (2) Regardless of whether or not the repeatability measures are skewed by the amount of population-level variation, declaring within-individual variation in arrival dates of ±14 days ‘consistent’ seems a bit odd. Many migratory birds vary their arrival dates by only ±3-5 days across years (again, see Conklin et al. 2013), and thus arrival dates that vary ±14 days do not appear especially consistent. I would therefore suggest reporting both measures of repeatability and within-individual variation in Table 1 (e.g., Arrival date for an individual ± variation across years for that individual).

Lines 285-287: “and the detours under these circumstances are even more difficult to explain” — this does not make sense since you go on to explain these detours in the rest of the Discussion.

Line 303: ‘Detoured’ is not the proper word here.

Lines 308-311: This is a run-on sentence.

Line 312: “might be one but not the only driver” — you need a comma between ‘one’ and ‘but.’

Lines 315-327: This addition seems tangential to your core focus and relies on data that you don’t fully present in this manuscript. I would leave it out.

Lines 355-359: See my above comments about repeatability and consistency.

---

## Round 0.3 · accepted · Accept

Thank you for addressing the comments raised in the second round of review, and improving the precision of language with respect to the terms 'consistency' and 'repeatability'. I am now happy to accept your manuscript for publication.